# Effects of Molecular Structure on the Physical Properties of Fully Substituted Cellulose Esters of Aliphatic Acids

**DOI:** 10.3390/polym17081053

**Published:** 2025-04-14

**Authors:** Taro Mori, Kanji Nagai, Shu Shimamoto

**Affiliations:** 1Biomass Innovation Center, R&D Headquarters, Daicel Corporation, Kakuma-machi, Kanazawa-shi 920-1192, Ishikawa, Japan; 2Graduate School of Natural Science and Technology, Kanazawa University, Kakuma-machi, Kanazawa-shi 920-1192, Ishikawa, Japan; 3Life Sciences R&D Center, PharmaTek BU, Life Sciences SBU, Arai Plant, Daicel Corporation, Myoko-shi 944-8550, Niigata, Japan; 4Business Development Center, R&D Headquarters, Daicel Corporation, Minato-ku, Tokyo 108-8230, Japan

**Keywords:** cellulose ester, mixed cellulose ester, thermal property, mechanical property, solubility parameter, property prediction

## Abstract

According to the literature, mainly in relation to mixed cellulose esters (MCEs) with two types of acyl group, the solubility parameter (SP) is a measure of certain physical properties of MCE that is helpful for property prediction; contrary to expectations, it has been implied by different studies that simple cellulose esters (SCEs) with only one type of acyl group do not altogether follow the empirical relationship. In this study, MCE and SCE were systematically prepared to verify the SP—property relationship. It was revealed that the correlation between SP and physical properties exists only for MCE. Thermal analysis revealed that MCEs possess remarkable unidentified endothermic transition depending on the ratio of two acyl groups, suggesting that the formation of such a stable structure could contribute to the difference between MCE and SCE. It was also revealed that, even for MCE, the empirical relationship involving SP varies from acyl group to acyl group; there is no universal relationship. In spite of aforementioned limitation, the empirical relationship with SP was verified to be useful for the property prediction of MCE. We demonstrated that the Fox equation and Voigt model are also useful for this prediction.

## 1. Introduction

According to the Organisation for Economic Co-operation and Development projections for 2060, there will be a substantial increase in the volume of global plastic waste driven by economic growth in developing countries. Global plastic consumption is expected to triple between 2019 and 2060 if no new policies are introduced [1]. By 2022, global plastic production was estimated to reach 400.3 million tons, with fossil-based plastics accounting for 362.3 million tons, or 90.6% of the total, while bio-based plastics contribute only 2.3 million tons (0.5% of the total) [2]. This context highlights the urgent need to shift toward renewable and potentially biodegradable bio-based plastics, given the finite nature of fossil resources and the environmental impacts associated with greenhouse gas (GHG) emissions during the production, disposal, and release of plastics into the environment [3,4]. Cellulose, the primary component of woody biomass, is the most abundant biomaterial on Earth, with an estimated annual production of 700 billion tons [5]. However, only a small fraction is utilized in cellulosic materials, such as regenerated cellulose and cellulose derivatives [6]. This limited usage is primarily due to cellulose’s strong intra- and intermolecular hydrogen bonding, which results in insolubility in most organic solvents and lack of melt processability [7].

Esterification, which uses the hydroxyl groups of cellulose as reactive sites, is a possible method for modifying the intrinsic properties of cellulose and introducing new functionalities, leading to the formation of cellulose esters (CEs) [8,9,10]. A prominent industrial example of esterification is cellulose acetate, where acetic acid is chemically bonded to cellulose via an ester linkage [11]. Cellulose acetate has become an indispensable industrial material, widely used in the production of fibers, cigarette filters, membranes, and various other applications owing to its unique properties [12]. However, cellulose acetate does not exhibit thermoplastic properties because it decomposes before melting [13]. Consequently, cellulose acetate resins have been developed by blending cellulose acetate with plasticizers [14,15]. Other commercially available CEs, such as cellulose acetate propionate and cellulose acetate butyrate, display a lower melt processability than petroleum-based plastics, necessitating the use of plasticizers in bioplastic applications [16]. Plasticizers incorporated into CEs, defined in this study as “external plasticizers,” generally have a low molecular weight, which poses limitations in the application of the resultant plastics. The addition of external plasticizers not only leads to leaching over time, resulting in the deterioration of mechanical properties [17], but also causes the release of volatile compounds that are harmful to humans and the environment during the molding process [18,19]. Chemical modification with flexible substituents on the cellulosic backbone has been implemented as an internal plasticization strategy to address issues associated with external plasticization [20,21]. Although simple CEs (SCEs) modified with long-chain aliphatic acids as internal plasticizers have been extensively studied [22,23,24,25], they have not been commercially utilized. From our point of view, this is due to two primary reasons: (1) an economical manufacturing process has yet to be established and (2) guidelines for designing CEs as plasticizer-free thermoplastics with the desired properties have not been fully developed. This study focused on the molecular design of such CEs.

Appropriately adjusting physical properties such as thermal stability and mechanical strength is essential when considering alternatives to petroleum-derived plastics. Mixed esterification of cellulose with both short and long alkyl moieties is an effective strategy for tuning the physical properties of CEs over a broad range because these properties are significantly influenced by the introduced substituents and the degree of substitution (DS) [26,27,28,29,30,31]. Edgar et al. investigated MCEs involving the two substituents, acetyl and acyl (propyl, butyl, hexyl, or nonanoyl) groups, and found liner correlations between the solubility parameter (SP) and the *T*_g_, logarithmic melt viscosity, or flexural modulus, drawing a conclusion that the properties of CE may be predicted with confidence without laboratory efforts [27]. The SP–physical property relationships established in the literature are indeed useful within the materials studied there. However, before extending the empirical relationship to any CEs, there is a serious question to address; as far as the melting points of a series of SCEs from other studies [22,23] are concerned, SCEs do not follow an empirical linear relationship involving SP (Figure 1).

In this study, we aim to investigate whether or not MCE and SCE are different in terms of SP–physical property relationships and to discuss better property predictions. To this end, a series of SCEs and MCEs were systematically prepared for the evaluation of physical properties, while the corresponding SP values were calculated by means of group contribution methods established by a few different groups with a view to assessing empirical relationships involving SP.

## 2. Materials and Methods

### 2.1. Materials

Lyocell fibers (Tencel, Lenzing AG, Lensing, Austria) cut into 5 mm long pieces were used as the starting material for the preparation of CEs. Lithium chloride (LiCl; 99%), *N*,*N*-dimethylacetamide (DMAc; ultra-hydrated, 99.5%), tetrahydrofuran (THF; 99.5%), and methanol (99.8%) were obtained from FUJIFILM Wako Pure Chemical Corporatio, Osaka, Japan. 4-Dimethylaminopyridine (DMAP; 99%), acetyl chloride (98%), hexanoyl chloride (98%), lauroyl chloride (98%), myristoyl chloride (98%), palmitoyl chloride (97%), and stearoyl chloride (96%) were sourced from Tokyo Chemical Industry Co., Ltd., Tokyo, Japan. All raw materials and solvents were used as received, without further purification or treatment.

### 2.2. Preparation of SCEs

The reaction is illustrated in Figure 1 and preparation parameters such as total acid chlorides, catalyst level, cellulose consistency, temperature, and time were determined considering previous works in relation to similar reaction systems [33,34,35,36,37,38,39]. First, 0.40 g (2.5 mmol, based on anhydrous glucose units) of cellulose (Tencel) was weighed into a test tube. Next, 15.04 g of dehydrated DMAc was added under a nitrogen atmosphere. The mixture was heated to 130 °C with continuous stirring for 2 h. Next, the mixture was allowed to cool to 25 °C, and 1.07 g of LiCl was added. The temperature then increased to 150 °C at a controlled rate of 20 °C every 10 min. The mixture was held at 150 °C for an additional 20 min. After confirming complete dissolution, 0.90 g (7.4 mmol) of DMAP was added, and the temperature was raised to 80 °C. Once DMAP was fully dissolved, acid chloride (22.5 mmol, corresponding to the target SCE) was added dropwise. The reaction proceeded at 80 °C for 3 h to ensure complete acylation. Methanol was then added to quench any unreacted acyl chloride. A substantial volume of methanol was subsequently added to precipitate the product, which was then filtered and washed with methanol to remove impurities. For further purification, the obtained product was dissolved in THF and re-precipitated by adding methanol. The resultant solid was collected by filtration, washed again with methanol, and vacuum-dried at 50 °C overnight. This process yielded the target SCEs, successfully completing their preparation.

### 2.3. Preparation of MCEs

A homogeneous solution was prepared following the same method as that used for SCEs. Before the addition of acid chlorides, acetyl chloride and lauroyl chloride were combined at molar ratios of 7.5/1.5, 6.0/3.0, 4.5/4.5, 3.0/6.0, and 1.5/7.5, totaling 22.2 mmol. This mixture was then added dropwise and allowed to react at 80 °C for 3 h. After the reaction was complete, methanol was added to quench any unreacted acyl chloride. A substantial volume of methanol or methanol aqueous solution was subsequently added to precipitate the product. The following steps in the preparation of SCEs were performed, resulting in the isolation of MCEs.

### 2.4. Chemical Characterization

A total of 10 mg of CE was dissolved in 1 g of deuterated chloroform (CDCl_3_) or deuterated dimethyl sulfoxide (DMSO-*d*_6_) to determine the DS, and ^1^H nuclear magnetic resonance (NMR) spectroscopy was conducted at 40 °C for CDCl_3_ or 80 °C for DMSO-*d*_6_ using a JNM-ECZL600R FT NMR spectrometer (JEOL Ltd., Tokyo, Japan). The DS was calculated using an integration method. The molecular weights of the CEs were measured by gel permeation chromatography (GPC) using a Prominence GPC system (Shimadzu Corporation, Kyoto, Japan) equipped with an RID-20A refractive index detector and two LF-804 columns (8.0 mm × 30 cm, Shodex, Tokyo, Japan). The eluent was either THF without a stabilizer or 20 mM LiBr/DMAc, with polystyrene used as the standard. GPC was performed at an oven temperature of 40 °C in THF or 50 °C in 20 mM LiBr/DMAc.

### 2.5. Thermal Properties

The thermal decomposition of the CEs was analyzed by thermogravimetric differential thermal analysis (TG-DTA) under a nitrogen atmosphere using a NEXTA STA200RV thermogravimetric analyzer (Hitachi High-Tech Analysis Corporation, Tokyo, Japan). The samples were heated from 30 to 550 °C at a rate of 20 °C/min, and the temperature at 5% weight loss (*T*_d5%_) was used as an index of thermal decomposition.

The glass transition temperature (*T*_g_) was determined by differential scanning calorimetry (DSC) under a nitrogen atmosphere using a NEXTA DSC600 differential scanning calorimeter (Hitachi High-Tech Analysis Corporation, Tokyo, Japan). The samples were first heated from room temperature to 240 °C at a rate of 10 °C/min and held at 240 °C for 5 min (first heating scan), followed by cooling from 240 to −80 °C at the same rate. *T*_g_ was estimated from the thermogram of the second heating scan, obtained by reheating the samples to 240 °C at the same rate.

### 2.6. Preparation of Hot-Pressed Films

The hot-pressed films of the CEs were prepared using an MP-SNL mini test press (Toyo Seiki Seisaku-sho, Ltd., Tokyo, Japan). The press molding of the CEs was performed at 200 °C under a pressure of 15 MPa between aluminum sheets, and the pressed films were rapidly cooled by flushing the press plates with water at 25 °C after heating. The films had an area of 25 cm^2^ and an average thickness of 500 μm.

### 2.7. Mechanical Properties

The mechanical properties of the CEs were measured using a Tensilon RTF-1350 system (A&D Company, Ltd., Tokyo, Japan) under conditions of 23 °C and 50% relative humidity. The tensile tests were conducted on the hot-pressed films processed into a dumbbell shape (film thickness = 500 μm, initial gauge length = 20 mm), with more than ten specimens tested for each CE. The tensile modulus was calculated from the initial linear slope of the stress–strain curve obtained at a tensile speed of 5 mm/min according to relevant ISO [40].

## 3. Results and Discussion

### 3.1. Esterification of Cellulose and Determination of DS

Table 1 enumerates representative samples of SCE and MCE systematically prepared in this study. The sample preparation was systematic in terms of the selection of acyl groups and substituent ratio of MCE. The preparation parameters were determined with reference to the literature [33,34,35,36,37,38,39]. The parameter settings for sample preparations were the same except for the type of acylation agent. As shown in Table 1, the total degree of substitution (DS_total_) was 3.0 for the samples. The apparent weight-average degree of polymerization (DP_w_) expressed as the polystyrene standard decreased as the carbon number increased. Malm et al. reported that, for a series of SCEs with a similar absolute degree of polymerization, the apparent intrinsic viscosity (limiting viscosity number) decreases as the carbon number increases [41]. Therefore, the systematic difference in SCEs in apparent DP_w_ observed in this study must be smaller or negligible if expressed in absolute DP_w_. The DS of the CEs was determined by ^1^H NMR spectroscopy. As an example, the NMR spectra of the CL3.0 (DS_La_3.0) and CAL1.6 (DS_La_1.6) are shown in Figure 2. The degree of substitution by the lauroyl group (DS_La_) was determined from the peak observed at 0.8–1.0 ppm, corresponding to the terminal methyl group (Equation (1)). The degree of substitution by the acetyl group (DS_Ac_) was calculated using Equation (2) because of the overlap of the acetyl peak (peak f in Figure 2B).(1)$${DS}_{La}=\frac{7.06\times I_{e(\delta 0.88)}}{3.02\times I_{a(\delta 3.0-5.5)}}$$(2)$${DS}_{Ac}=\frac{7.06\times I_{f,b(\delta 1.8-2.5)}-{2.06\times {DS}_{La}\times I}_{a(\delta 3.0-5.5)}}{3.02\times I_{a(\delta 3.0-5.5)}}.$$

The DS, molar mass, and degree of polymerization of the SCEs and MCEs prepared in this study are listed in Table 1. In Table 1, DS_x_/DS_Total_ is defined as the substituent ratio, representing the fraction of the longer substituent. Further data regarding the preparation are shown in Appendix A.

### 3.2. Thermal Properties of MCEs

All of the CALs were successfully molded into hot-pressed films and strip-shaped plates at 200 °C without any external plasticizer incorporated, suggesting the potential of CALs as thermoplastics. The thermal decomposition and transition temperatures of the SCEs and CALs were evaluated by TG-DTA and DSC. The 5% weight loss temperatures (*T*_d5%_), transition temperatures (*T*_m_, *T*_g_, and *T*_p_), and the corresponding enthalpy changes (Δ*H*_m_ and Δ*H*_p_) are summarized in Table 2. The thermogravimetric analysis (TGA) profiles obtained by heating these CEs from 30 °C to approximately 500 °C at a rate of 20 °C/min are shown in Appendix A. Previous research indicates that the thermal decomposition of SCEs is influenced by the degree of acyl substitution, with higher substitution levels contributing to relatively high thermal stability [42]. In contrast, the carbon number of the acyl group has minimal effect on the thermal stability [43]. The TG-DTA results reveal that the primary decomposition of the CALs begins at approximately 320 °C, with no significant difference in thermal stability observed in relation to that of the SCEs.

Figure 3 shows the second-heating DSC thermograms of the CEs. The thermograms display a broad glass transition for all of the CEs and, in some cases, one or two broad endothermic peaks (*T*_m_ and *T*_p_). Among these thermal transitions, the low-temperature endothermic peaks (*T*_m_) are associated with the melting of the acyl groups. Figure 3A shows that SCEs with C12 or a longer acyl chain exhibit a peak corresponding to the melting point of the aliphatic acid groups in the DSC thermogram, as previously reported [25]. For these SCEs, the highest *T*_m_ and largest Δ*H*_m_ were observed for the stearoyl group, which had the longest alkyl chain, and as the carbon number decreased, the *T*_m_ shifted to lower values and the Δ*H*_m_ decreased. In Figure 3B, the peaks of *T*_m_ were observed in the low-temperature regions of CL3.0, CAL2.6, and CAL2.1. Furthermore, the Δ*H*_m_ of these peaks tended to decrease with a reduction in the lauroyl substituent ratio, and the *T*_m_ was absent for CAL1.6. The disappearance of the *T*_m_ observed for the CALs suggests that the crystallization of lauroyl groups is inhibited by modification with acetyl groups. The *T*_p_ is an endothermic peak observed at a temperature higher than the *T*_g_ by approximately ~40 °C, which has not been attributed to any type of transition in this study. Similar endothermic peaks were reported for similar CEs in previous studies, but in our view, the origin of such endothermic peaks is still controversial; Glasser et al. reported that they were melting points [26], while Yamagishi et al. and Tanaka et al. independently reported that they could be a transition from an anisotropic to isotropic phase [29,44]. The *T*_p_ must be some kind of a transition but could not be attributed to either the melting point or anisotropic-to-isotropic-phase transition in our study. The *T*_g_ values of CEs were influenced by the DS and the carbon number of the acyl group. Previous studies have indicated that a higher DS or a longer acyl chain (up to around C7) lowers the *T*_g_ [45,46], whereas acyl chains with C12 or more carbon atoms slightly increase the *T*_g_ due to crystallization interactions between the acyl groups [25]. For the CALs, *T*_g_ values were observed between CA3.0 and CL3.0, with an interpolative trend in which increasing the lauroyl substituent ratio led to a decrease in the *T*_g_ of the CALs.

### 3.3. SP–Thermal Properties Relationship of CEs

Figure 4A shows the *T*_g_ values of the SCEs and CALs as a function of the carbon number of substituents; for CALs, the carbon number represents the average for two substituents taking into account the degree of substitutions. In the case of SCEs, the relationship between carbon number and *T*_g_ is not simple, showing a minimum of *T*_g_ at the carbon number 6 (sample CH3.0). The observed behavior of *T*_g_ of the SCEs can be regarded as similar in nature to that of the melting point (Figure 1A). On the other hand, in the case of the CALs, the *T*_g_ values were roughly expressed by interpolation of the *T*_g_ values of the corresponding SCEs, CA3.0 and CL3.0. Figure 4B shows the correlation between the *T*_g_ and SP of the CEs; the SP values are calculated from the method after Coleman et al. [32]. For the SCEs, a minimum of *T*_g_ was observed at around a SP value of 9.2 (cal/cm^3^)^1/2^. On the other hand, in the case of the CALs, the relationship between the *T*_g_ and SP is monotonic; the higher the SP, the higher the *T*_g_ as reported for cellulose acetate nonanoate and MCEs with shorter acyl chains by Edgar et al. [27]. From Figure 4A, it was revealed that the MCEs tend to possess higher *T*_g_ values than the SCEs, suggesting that the MCEs form more stable solid structures than the SCEs. Considering that the chemical structures of fully substituted SCEs possess symmetries subject to crystallization, whereas the heterogeneity in chemical structures of MCEs does not enable them to crystallize, it is somehow counterintuitive that the solid structures of MCEs are more stable than that of SCEs. Further efforts will be required to elucidate what makes the solid structures of MCEs stable. Watanabe et al. reported that heterogeneity in the chemical structure of synthetic peptide promotes liquid crystallinity [47]. Microscopic observations in this study for the CALs and SECs did not show any sign of the formation of thermotropic liquid crystals. DCS studies, however, revealed that the Δ*H*_p_ values of unidentified endothermic peaks (*T*_p_) of the CALs were larger than that of SCEs, showing the maximum when the substitution ratio was 0.5 (the balance of acetyl and lauroyl groups was approximately 1:1), meaning that some kind of stable structure was formed at the substituent ratio (Table 2). The formation of such a stable structure could contribute to the difference between MCE and SCE.

### 3.4. Prediction of the Physical Properties of MCEs

From the SP–thermal property relationship for CALs, as shown in Figure 4B, it is demonstrated that the *T*_g_ of CALs changed linearly as a function of the SP, making it possible to predict the *T*_g_ based on the SP as reported by Edgar et al. [27]. However, even for MCE, the empirical relationship with the SP varied from acyl group to acyl group, and no universal relationship or master curve exists as a function of the SP. Here, we examined the relationship between the SP and mechanical properties of CALs using the same approach as for *T*_g_. Additionally, we also investigated the correlation between the SP calculated from other group contribution methods established by a few different groups and the physical properties of MCEs. Tensile tests were conducted on the hot-pressed CL3.0 and CALs films to evaluate the effect of the substituent ratios of the MCEs on mechanical properties. CAL0.5, CAL1.6, and CAL2.4 were prepared again using the same method as described above. Appendix A lists the DS, molar mass, and degree of polymerization. Table 3 summarizes the mechanical properties of these samples, including the tensile strength, elongation at break, and tensile modulus. The plots of these mechanical properties as function of the substituent ratio and stress–strain curves are shown in Appendix A. The results of this experiment indicated that the mechanical properties measured by the tensile test correlated with the lauroyl substituent ratios of the CALs. As the substituent ratios increased, both the tensile strength and tensile modulus decreased, while elongation at the break increased.

We investigated the relationships of the *T*_g_ and mechanical properties of CALs with the SP calculated from the methods of Coleman [32], Fedors [48], and Hoftyzer-Van Krevelen [49]. Figure 5A shows the *T*_g_ of the CA3.0, CL3.0, and CALs as a function of the SP calculated by these three methods. Although the slopes of the linear approximations naturally varied depending on the calculation methods of SP, all the correlations exhibited a similar linear trend. Figure 5B shows the tensile strength of the CL3.0 and CALs as a function of the SP. The tensile strength exhibited a strong linear correlation regardless of which methods were used for calculating the SP. The tensile modulus also correlated linearly with the SP, as shown in Appendix A. As reported by Edgar et al. for the MCEs with C9 or shorter acyl chains, we demonstrated that the *T*_g_ and mechanical properties of CALs, unlike the case of the SCEs with the long acyl groups, could be predicted based on the liner relationship involving SP.

We also examined whether the *T*_g_ and mechanical properties of CALs could be predicted from other methods using a model equation based only on the physical properties of the corresponding SCEs, CA3.0 and CL3.0 without using the SP. For *T*_g_, empirical evidence suggests that the experimental *T*_g_ values of copolymers are typically in good agreement with the Fox equation [50,51,52]. In the case of CEs, the *T*_g_ of cellulose acetate with external plasticizers and partially substituted SCEs with long acyl chains as internal plasticizers aligns well with the *T*_g_ calculated by the Fox equation for a mixture of cellulose acetate and an external plasticizer or unsubstituted cellulose and eicosane (C20 alkane) [53,54,55]. Considering the CALs as copolymers of CA3.0 and CL3.0, we compared the *T*_g_ values calculated using the Fox equation with the experimental values. Figure 6 shows the experimental *T*_g_ (data points) and calculated *T*_g_ (dashed line) as a function of the lauroyl substituent ratios. The calculated values were obtained from the Fox equation (Equation (3)) using the *T*_g_ values (*T*_gCA3.0_, *T*_gCL3.0_) of CA3.0 and CL3.0, with the weight fractions (*w*_CA3.0_, *w*_CL3.0_) corresponding to the substituent ratios of the acetyl and lauroyl groups in each CAL.(3)$$\frac{1}{T_{g}}=\frac{w_{CA3.0}}{T_{gCA3.0}}+\frac{w_{CL3.0}}{T_{gCL3.0}}$$

The experimental values for the CALs closely match the calculated values, suggesting that the *T*_g_ of MCEs can be predicted directly from only that of SCEs using the Fox equation.

Following the prediction of *T*_g_, we attempted to predict the mechanical properties of the CALs using model equations and compared the results with the experimental values. By treating the CALs as a mixture of the corresponding SCEs, CA3.0 and C3.0, we applied the Voigt model, one of the simplest models for binary mixtures known as the rule of mixtures [56,57,58], for predicting the mechanical properties of the CALs. Figure 7A, B illustrate the predicted curves of the tensile strength and elastic modulus for the CALs calculated using Equations (4) and (5), which represent the Voigt model, plotted alongside the experimental values. The molar volume fractions (*V*_CA3.0_ and *V*_CL3.0_), elastic moduli (*E*_CA3.0_ and *E*_CL3.0_), and tensile strengths (*σ*_RCA3.0_ and *σ*_RCL3.0_) in the equations are based on the substituent ratios of the acetyl and lauroyl groups in each CAL.(4)$$E=V_{CA3.0}\times E_{CA3.0}+V_{CL3.0}\times E_{CL3.0}$$(5)$$\sigma_{R}=V_{CA3.0}\times \sigma_{RCA3.0}+V_{CL3.0}\times \sigma_{RCL3.0}$$

The values of tensile strength and elastic modulus of CA3.0 used for this prediction are taken from the literature [22,59], while that of CL3.0 are experimental data. The good agreement between the observed and calculated values suggests that it is possible to predict the mechanical properties of MCEs directly from only the property values of the corresponding SCEs using the Voigt model.

As previously mentioned, it is important to note that the relationship between SP and physical properties is not universal across all MCEs. The relationship provides a linear prediction by means of interpolation for the corresponding SCEs. Table 4 summarizes the parameters for the prediction of *T*_g_ for MCEs by utilizing the SP and Fox equation, respectively. Upon comparing the root mean squared errors (RMSEs) associated with the prediction methods for the Tg of CALs, the accuracy of the prediction using the Fox equation was larager than that of the method based on the SP. Both methods should be applied with the limitation in accuracy. The parameters, including the literature value in Malm et al. [22] for the prediction of tensile strength, are summarized in Table 5 in the same manner as for *T*_g_. The values of RMSE were 3.0 and 3.5 calculated from the SP and Voight model, respectively. Both methods should be applied with the limitation in accuracy. Using the parameters presented in Table 4 and Table 5 makes it possible to predict the *T*_g_ and tensile property of the MCEs with various substituent ratios of the acetyl and acyl groups (C6, C12, C14, C16, or C18) without laboratory experiments.

## 4. Conclusions

The correlation between SP and physical properties exists only for MCE. There is no such correlation for SCE. The origin of such difference between MCE and SCE has yet to be further studied.Even for MCE, the empirical relationship involving SP varies from acyl group to acyl group; there is no universal relationship.In spite of the aforementioned limitation, the empirical relationship with SP is useful for the property prediction of MCE as long as acyl groups are specified. The Fox equation and Voigt model could also be utilized for the prediction based only on the properties of the corresponding SCEs. The observed root mean squared errors for prediction were 8.0 °C (*T*_g_ based on SP), 13.2 °C (*T*_g_ based on Fox equation), 3.0 MPa (tensile strength based on SP), and 3.5 MPa (tensile strength based on Voigt model).

## Data Availability

The original contributions presented in this study are included in the article/Appendix A. Further inquiries can be directed to the corresponding author.

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
