# Peer review of "Effects of Molecular Structure on the Physical Properties of Fully Substituted Cellulose Esters of Aliphatic Acids"

_polymers, 2025, doi:10.3390/polym17081053_

Round 1

Reviewer 1 Report

Comments and Suggestions for Authors

In this manuscript, the authors synthesized a series of fully substituted cellulose acetate laurate with systematically varied substituent ratios of acetyl and lauroyl groups, as well as fully substituted SCEs modified with different alkyl length. Thermal, viscoelastic, and mechanical properties were investigated and compared. Overall, some reasonable results were obtained. Revisions should be made before accepted.

1. The full name of "OECD" should be given when mentioned for the first time at the beginning of Introduction section.

2. What is the meaning of Tp? More information should be provided and discussed for Tp.

3. There are a couple of grammar mistakes in the manuscript, please check carefully.

Comments on the Quality of English Language

The English language can be further improved.

Author Response

Thank you very much for your kind comments. Taking them into account, we have revised our manuscript as you see below.

Comments 1: The full name of "OECD" should be given when mentioned for the first time at the beginning of Introduction section.

Response 1: We appreciate the comment No. 1 of Reviewer 1. The manuscript is now modified accordingly. Please refer to lines 31 of the manuscript.

Comments 2: What is the meaning of Tp? More information should be provided and discussed for Tp.

Response 2: We appreciate the comment No. 2 of Reviewer 1. The manuscript is now modified accordingly. Please refer to lines 238 to 246 of the manuscript.

Comments 3: There are a couple of grammar mistakes in the manuscript, please check carefully.

Response 3: We appreciate the comment No. 3 of Reviewer 1. English Grammar of the manuscript be double checked by the service provided by MDPI. I would like to propose that such an English Grammar check will take place once the review process is finalized as long as acceptable to you.

Reviewer 2 Report

Comments and Suggestions for Authors

1. After reading the whole introduction, the significance and novelty of this work are still not clear. Please modify the relevant content.

2.  If possible, please provide design basis of preparation parameters for MCE.

3. Please indicate the specific application scenarios of this work.

4. Authors should point out breakthrough progress of this work comparing with other related literature. 

5. Conclusions should contain detailed quantitative technical indicators and be described systematically.

Author Response

Thank you very much for taking the time to review this manuscript. Taking them into account, we have revised our manuscript as you see below.

Comments 1: After reading the whole introduction, the significance and novelty of this work are still not clear. Please modify the relevant content.

Response 1: We appreciate the comment No. 1 of Reviewer 2. The manuscript is now modified accordingly. Please refer to lines 77 to 95 of the manuscript and Figure 1.

Comments 2: If possible, please provide design basis of preparation parameters for MCE.

Response 2: We appreciate the comment No. 2 of Reviewer 2. The manuscript is now modified accordingly. Please refer to lines 107 to 109 of the section 2.2. Preparation of SCEs and lines 175 to 185 of the section 3.1. Esterification of Cellulose and Determination of DS in the manuscript.

Comments 3: Please indicate the specific application scenarios of this work.

Response 3: We appreciate the comment No. 3 of Reviewer 2. The manuscript is now modified accordingly. Please refer to Table 4 and Table 5 and the relevant text (Lines 376 to 388).

Comments 4: Authors should point out breakthrough progress of this work comparing with other related literature. 

Response 4: We appreciate the comment No. 4 of Reviewer 2. The manuscript is now modified accordingly. Please refer to the revised Abstract in the manuscript.

Comments 5: Conclusions should contain detailed quantitative technical indicators and be described systematically.

Response 5: We appreciate the comment No. 5 of Reviewer 2. Conclusions are now modified accordingly. Please refer to the revised Conclusions in the manuscript.

Reviewer 3 Report

Comments and Suggestions for Authors

The work conducted by Mori T. and co-authors aims to establish guidelines for designing plasticizer-free CEs suitable as thermoplastics based on fully substituted mixed cellulose esters with both short and long alkyl moieties. The authors focused their study on cellulose acetate laurates with varying acetyl-to-lauroyl ratios and fully substituted SCEs modified with acetyl (denoted as C2), hexanoyl (C6), lauroyl (C12), myristoyl (C14), palmitoyl (C16) and stearoyl (C18) groups. In my opinion, the presented manuscript is of sufficient novelty and is well-written. The resulting materials were characterised in terms of structural, thermal, processing and mechanical behaviour. The material's characterisation and discussion are comprehensive and detailed.

However, I have several comments the authors might consider:

1.      Tensile tests methodology - How tensile modulus was calculated? What type of extensometer was used in this study? Usually, Tensile modulus is obtained from the initial - linear slope of the stress-strain curve when the specimen is stretched at 1%/min. What was the reason for taking an unconventional approach (5%/min) in this case?

2.      Thermal properties – in this section, in addition to Tm and Tg, the authors discuss Tp - the endothermic transition that is observed on the heating profile at temperatures above the glass transition temperature - What is this transition? can it be identified on the heating curve (please indicate), and what effects are associated with this transition? 

3.      Mechanical properties of MCEs - Can exemplary representative stress-strain curves be provided? Readers may be interested in what kind of character they exhibit. I believe that adding exemplary curves may bring new information and broaden the discussion.

Additionally, there are some minor remarks:

1.      Introduction - “OECD projections” – all acronyms used in the text should be initially presented in their complete form; thus, it should be “Organisation for Economic Co-operation and Development projections”.

2.      Introduction – “no melt processability” (Line 48) can be replaced with “lack of melt processability” or “nonprocessable by melting”.

Comments on the Quality of English Language

In some points the English could be improved to more clearly express the research.

Author Response

Thank you very much for your kind comments. Taking them into account, we have revised our manuscript as you see below.

Comments 1-1: Tensile tests methodology - How tensile modulus was calculated?

Response 1-1: In this study, the tensile modulus was calculated from the initial linear slope of the stress-strain curve obtained at a tensile speed of 5 mm/min. We have added the relevant text and please refer to lines 171 to 172 of the revised manuscript.

Comments 1-2: What type of extensometer was used in this study?

Response 1-2: Tensilon RTF-1350 (A&D Company, Ltd.) was used as a contact extensometer (lines 167 to 168 of the manuscript).

Comments 1-3: Usually, Tensile modulus is obtained from the initial - linear slope of the stress-strain curve when the specimen is stretched at 1%/min. What was the reason for taking an unconventional approach (5%/min) in this case?

Response 1-3: As the reviewer pointed out, specimens with a shape of dumbbell specified in ISO 527 [Plastics - Determination of tensile properties] such as type 1A and 1B are usually tested at a tensile speed of 1%/min. In this study, tensile tests were measured For film specimens, tensile speeds in previous studies vary from author to author. Crépy et al. employed a tensile speed of 1.44 mm/min. [Carbohydr. Polym. 2020, 234, 115912]. Willberg-Keyriläinen et al. employed a tensile speed of 5 mm/min. [Cellulose 201724, 505–517.] Danjo et al. empolyed a tensile speed of 20 mm/min. [Polymer 2018, 137, 358–363.]. Considering previous studies as such, 5 mm/min was employed in this study as the minimum speed allowed specifically to a sample with film form in ISO527-3 [Part 3: Test conditions for films and sheets].

Comments 2: Thermal properties – in this section, in addition to Tm and Tg, the authors discuss Tp - the endothermic transition that is observed on the heating profile at temperatures above the glass transition temperature - What is this transition?

Response 2: We appreciate the comment No. 2 of Reviewer 3. The manuscript is now modified accordingly. Please refer to Figure 3 and lines 238 to 246 of the section 3.2. Thermal Properties of MCEs and lines 256 to 282 of the section 3.3. SP – thermal properties relationship of CEs in the manuscript.

Comments 3: Mechanical properties of MCEs - Can exemplary representative stress-strain curves be provided? Readers may be interested in what kind of character they exhibit. I believe that adding exemplary curves may bring new information and broaden the discussion.

Response 3: We appreciate the comment No. 3 of Reviewer 3. The manuscript is now modified accordingly. Please refer to Figure S3 of the supplementary material.

Comments 4: Introduction - “OECD projections” – all acronyms used in the text should be initially presented in their complete form; thus, it should be “Organisation for Economic Co-operation and Development projections”.

Response 4: We appreciate the comment No. 4 of Reviewer 3. The manuscript is now modified accordingly. Please refer to lines 31 of the manuscript.

Comments 5: Introduction – “no melt processability” (Line 48) can be replaced with “lack of melt processability” or “nonprocessable by melting”.

Response 5: We appreciate the comment No. 5 of Reviewer 3. The manuscript is now modified accordingly. Please refer to lines 46 of the manuscript.

Round 2

Reviewer 2 Report

Comments and Suggestions for Authors

The manuscript has been carefully revised and can be accepted.